



# Determination of High-Precision Tropospheric Delays Using Crowdsourced Smartphone GNSS Data

Yuanxin Pan[1], Grzegorz Kłopotek[1], Laura Crocetti[1], Rudi Weinacker[2], Tobias Sturn[2], Linda See[2], Galina Dick[3], Gregor Möller[4], Markus Rothacher[1], Ian McCallum[2], Vicente Navarro[5], and Benedikt Soja[1]

[1]Institute of Geodesy and Photogrammetry, ETH Zurich, Zurich, Switzerland
[2]International Institute for Applied Systems Analysis (IIASA), Laxenburg, Austria
[3]GFZ German Research Centre for Geosciences, Potsdam, Germany
[4]Department of Geodesy and Geoinformation, TU Wien, Vienna, Austria
[5]European Space Agency, European Space Astronomy Center, Madrid, Spain

**Correspondence:** Yuanxin Pan (yxpan@ethz.ch)

**Abstract.** The Global Navigation Satellite System (GNSS) is a key asset for tropospheric monitoring. Currently, GNSS meteorology relies primarily on geodetic-grade stations. However, such stations are too costly to be densely deployed, which limits the contribution of GNSS to tropospheric monitoring. In 2016, Google released the raw GNSS measurement application programming interface for smartphones running on Android version 7.0 and higher. Since nowadays there are billions of Android

smartphones worldwide, utilizing those devices for atmospheric monitoring represents a remarkable scientific opportunity. In this study, smartphone GNSS data collected in Germany as part of the Application of Machine Learning Technology for GNSS IoT Data Fusion (CAMALIOT) crowdsourcing campaign in 2022 were utilized to investigate this idea. Approximately twenty thousand raw GNSS observation files were collected there during the campaign. First, a dedicated data processing pipeline was established that consists of two major parts: machine learning (ML)-based data selection and ionosphere-free Precise Point Po-

sitioning (PPP)-based Zenith Total Delay (ZTD) estimation. The proposed method was validated with a dedicated smartphone data collection experiment conducted on the rooftop of the ETH campus. The results confirmed that ZTD estimates of mm-level precision could be achieved with smartphone data collected in an open-sky environment. The impacts of observation time span and utilization of multi-GNSS observations on ZTD estimation were also investigated. Subsequently, the crowdsourced data from Germany were processed by PPP with the ionospheric delays interpolated using observations from surrounding SAPOS

(Satellite Positioning Service of the German State Survey) GNSS stations. The ZTDs derived from ERA5 and an ML-based ZTD product served as benchmarks. The results revealed that an accuracy of better than 10 mm can be achieved by utilizing selected high-quality crowdsourced smartphone data. This study marks the first successful demonstration of high-precision ZTD determination with crowdsourced smartphone GNSS data and reveals success factors and current limitations.

## 1 Introduction

The Global Navigation Satellite System (GNSS) provides the capability to continuously monitor the troposphere with high precision, regardless of weather conditions (Hofmann-Wellenhof et al., 2012). One of the primary meteorological parameters





derived from GNSS observations is Zenith Total Delay (ZTD), which can be further converted to Precipitable Water Vapor (PWV). Since water vapor is a highly dynamic meteorological variable, exhibiting significant spatial and temporal variations, it necessitates regular measurement with dedicated sensors. However, the current meteorological observing system lacks suffi-

ciently dense measurements. Hence, the increased utilization of GNSS observations can make substantial contributions to both meteorology and climatology studies (Bevis et al., 1992).

Currently, GNSS meteorology mainly relies on the data obtained from geodetic-grade receivers of global or regional networks, such as the widely recognized International GNSS Service (IGS) network and the European E-GVAP (EUMETNET EIG GNSS Water Vapour Programme) network (Guerova et al., 2016). Typically, ZTDs derived from those geodetic-grade sta-

tions exhibit an accuracy of several millimeters (Li et al., 2015). These precise ZTD estimates can be further assimilated into Numerical Weather Models (NWM), thereby enhancing the accuracy of weather forecasting (Guerova et al., 2016). However, geodetic-grade GNSS receivers are costly and cannot be densely deployed, especially in less developed regions, which limits their contribution to meteorology research.

The potential of utilizing cost-effective GNSS devices for tropospheric monitoring has been discussed in many publications.

Single-frequency GNSS receivers offer an alternative solution in this regard. Wang et al. (2019) examined the performance of single-frequency GNSS stations for relative ZTD estimation and found that they could achieve a precision comparable to that derived from dual-frequency stations. Krietemeyer et al. (2018) adopted the Satellite-specific Epoch-differenced Ionospheric Delay (SEID) method (Deng et al., 2009) to recover the second-frequency observations for single-frequency receivers, and achieved a ZTD estimation accuracy of 4 mm using Precise Point Positioning (PPP). Stępniak and Paziewski (2022) evaluated

ZTDs derived from u-blox receivers and reported mm-level agreement with those derived from geodetic-grade GNSS receivers. They also highlighted the crucial role of GNSS antennas in achieving high-precision ZTD retrieval and suggested that the utilization of a geodetic-grade antenna could further improve the results.

Given that there are around 3 billion Android smartphones being used worldwide (Cranz, 2021), they represent a source of ubiquitous low-cost GNSS devices. Raw GNSS data from smartphones have become accessible from 2016 onward after

Google released the corresponding Application Programming Interface (API) for the Android 7.0 operating system (Banville and Van Diggelen, 2016). Subsequently, more advanced positioning algorithms have been developed to exploit the raw GNSS data. However, studies have also revealed several issues with these smartphone GNSS data, including weak resistance to multipath interference, the frequent occurrences of cycle slips, and the unreliability of second-frequency observations (Zhang et al., 2018; Li and Geng, 2019). These issues are attributed to the passive patch antennas and low-cost GNSS chips commonly

employed in smartphones, and thus make it challenging to use smartphone data for tropospheric monitoring.

Currently, research in relation to smartphone-based GNSS observations focuses mainly on the analysis of data quality and the development of advanced positioning algorithms (Paziewski, 2020; Zangenehnejad and Gao, 2021). However, the subject of smartphone-based tropospheric monitoring remains relatively scarce. Tagliaferro et al. (2019) presented initial results of ZTD estimation using a Nexus 9 tablet and a Xiaomi 8 smartphone. They employed the SEID method to recover dual-frequency

observations and achieved an accuracy of about 5 mm for ZTD estimation compared to those derived from a nearby geodetic-grade receiver. Benvenuto et al. (2021) also explored ZTD estimation with smartphones and reported an accuracy of several



centimeters. In another study, Stauffer et al. (2023) tested relative ZTD estimation using two weeks of data collected by a Google Pixel 4XL device and demonstrated that an accuracy of better than 10 mm could be achieved. However, it is worth noting that these studies only used a few dedicated smartphone GNSS data sets, and the potential of harnessing massive smartphone GNSS observations remains largely uncharted. One of the main challenges has been how to collect the data at scale, apart from developing dedicated methods to select GNSS data of sufficient quality and utilizing a suitable approach to process such diverse observations.

Crowdsourcing has proven to be a valuable tool for data collection in scientific research (Clery, 2011; See et al., 2022). Its effectiveness is well demonstrated in applications such as earthquake early warning using crowdsourced smartphone acceleration measurements (Kong et al., 2016; Allen et al., 2020). However, the use of crowdsourced smartphone GNSS data for tropospheric monitoring has remained unexplored until now. Prior research by Marques et al. (2021) and Lehtola et al. (2022) introduced a method for jointly estimating station positions and tropospheric delays using a crowdsourced smartphone network, yet their studies were limited to simulated GNSS data. To take advantage of the raw GNSS data API for Android smartphones and explore the potential of crowdsourced smartphone data for atmospheric monitoring, the Chair of Space Geodesy at ETH Zurich, in collaboration with the International Institute for Applied Systems Analysis (IIASA), launched the Application of Machine Learning Technology for GNSS IoT Data Fusion (CAMALIOT) crowdsourcing campaign in March 2022. A dedicated Android smartphone application, hereafter referred to as CAMALIOT app, has been developed and can be freely downloaded from the Google Play store. This app allows users to collect raw GNSS data for their own purposes and additionally lets them voluntarily upload the data to the CAMALIOT server. As a result of the conducted crowdsourcing campaign, around 12 thousand volunteers worldwide contributed over 5 TB of raw GNSS observations (See et al., 2023; Soja et al., 2023). An overview of the software architecture deployed on the CAMALIOT server, designed and implemented to handle collection and retention of the GNSS community data at scale, is given by Kłopotek et al. (2023).

In this study, we focus on the determination of high-precision tropospheric delays using the smartphone GNSS data collected during the CAMALIOT crowdsourcing campaign. The method concerning selection of suitable crowdsourced data as well as PPP-based ZTD estimation is given in Section 2. A dedicated smartphone data collection experiment conducted on the rooftop of the ETH campus and an overview of the crowdsourced data from Germany are described in Section 3. Subsequently, a detailed analysis of the data quality and an evaluation of smartphone-based ZTD estimation are provided in Section 4. Conclusions and outlook are given in Section 5.

## 2 Methodology

This section is dedicated to the established processing pipeline for crowdsourced smartphone GNSS data. An overview of the described approach is depicted in Figure 1. At a high level, the developed pipeline consists of two steps, where the first step is to select high-quality smartphone data in an automatic manner, whereas the second step concerns the estimation of tropospheric delays using the data selected in step one.





## 2.1 Data selection

While the crowdsourcing of data offers researchers the opportunity to access numerous and widespread observations at a relatively low cost, it has some limitations, with data quality being the primary concern. A substantial portion of the smartphone GNSS data collected during the CAMALIOT crowdsourcing campaign was low quality and could not be utilized for atmospheric monitoring (See et al., 2023), even though the CAMALIOT app provides guidance for GNSS data collection. However, it should be acknowledged that there is a clear trade-off between simplifying data collection efforts and the desire to collect

high-quality data. It means that users generally collected data regardless of observation environments and instances where smartphones were placed outdoors with an unobstructed view were rare. Consequently, data selection and quality control are essential for preprocessing the extensive volume of crowdsourced data. By excluding low-quality data, a considerable amount of computational resources can be saved during the subsequent data analysis.

To enable automatic data selection, a set of data quality indicators, e.g., Carrier-to-Noise Density ratio ($C/N_0$), Position

Dilution of Precision (PDOP), and observation noise, were initially extracted from the raw GNSS observations stored in the Receiver Independent Exchange format (RINEX) files. Then, a machine learning (ML)-based classifier was trained using a subset of the data that had already been manually labeled. During the crowdsourcing campaign, the developed ML-based classifier could automatically label incoming data as either 'Good' or 'Bad'. Therefore, smartphone data characterized by low quality were excluded from further processing. The developed ML-based classifier was characterized by precision and recall

scores of 0.96 and 0.97, respectively. Finally, around 0.7% of the data were classified as 'Good'. Detailed information on the training procedures and the performance evaluation of the ML-based classifier is described in Appendix A.

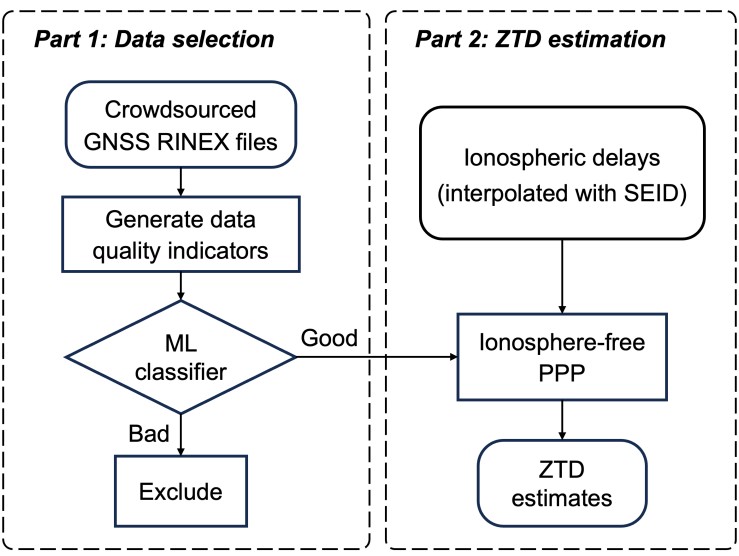

**Figure 1.** Flowchart of the data processing pipeline for crowdsourced smartphone GNSS data.





## 2.2 Estimation of zenith total delays

In this study, PPP is employed to estimate zenith total delays from raw GNSS data. PPP is a technique for determining the coordinates and other parameters of interest for a single receiver using high-precision satellite orbit and clock products (Zumberge et al., 1997; Kouba and Héroux, 2001). It is preferred over relative positioning because it does not require a reference station, especially considering that the crowdsourced data can be distributed across wide geographical areas and a suitable reference station cannot be always available. The fundamental observation equations for the GNSS pseudorange and carrier phase measurements are presented below:

$$P = \rho + c \cdot (d_r - d_s) + mf_h \cdot Z_h + mf_w \cdot Z_w + \theta$$
$$L = \rho + c \cdot (d_r - d_s) + mf_h \cdot Z_h + mf_w \cdot Z_w + \lambda \cdot N + \epsilon \tag{1}$$

where $P$ and $L$ are the ionosphere-free (IF) combinations of pseudorange and carrier phase measurements, expressed in units of meters; the geometric distance between the satellite and the station antenna phase center is denoted by $\rho$; the receiver and satellite clock biases are denoted by $d_r$ and $d_s$, respectively; the Zenith Hydrostatic Delay (ZHD) and Zenith Wet Delay (ZWD) are represented by $Z_h$ and $Z_w$, and their sum is the so-called ZTD; the mapping functions for ZHD and ZWD are denoted by $mf_h$ and $mf_w$, respectively; the unknown ambiguity is $N$ and its corresponding wavelength is $\lambda$; the observation noises of the ionosphere-free combinations of pseudorange and carrier phase measurements are denoted by $\theta$ and $\epsilon$, respectively.

Commonly, dual-frequency observations are needed to form ionosphere-free combinations and thus eliminate ionospheric delays. However, this remains a challenge for smartphone GNSS data. On the one hand, there are a limited number of smartphone models capable of recording dual-frequency observations. On the other hand, both the quantity and quality of observations on the second frequency cannot be always ensured for smartphones (Wang et al., 2020; Stauffer et al., 2023). Consequently, smartphone GNSS data usually cannot support IF-PPP, especially when data are crowdsourced from less favorable environments. To address this issue, we employed the SEID method to interpolate ionospheric delays at the smartphone locations using the surrounding geodetic GNSS stations. This allows to recover the second-frequency observations and use them in the GNSS analysis. The accuracy of this ionospheric delay interpolation method is typically accurate at the level of a few millimeters (Deng et al., 2009), especially during the inactive periods of the ionosphere. The Global Ionosphere Maps (GIM) (Schaer, 1999) were not used due to their limited precision.

The GNSS data were processed in static mode using our in-house software PPPx to derive ZTD estimates. The details of the PPP processing strategy is outlined in Table 1. Note that the receiver Phase Center Variation (PCV) corrections were not applied for smartphone GNSS observations due to the absence of smartphone GNSS antenna models. While GNSS data collected by smartphones are typically sampled at a 1-Hz rate, we decided to down-sample the data to 30-s intervals in order to enhance computational efficiency as such a sampling rate is still adequate for tropospheric monitoring purposes. Note that we utilized an elevation-dependent weighting scheme for the smartphone data processing. Although C/N$_0$-dependent weighting has proven to be advantageous in improving kinematic positioning precision (Zhang et al., 2018), our investigations revealed





that C/N$_0$-dependent weighting tends to introduce artifacts in ZTD estimates, and thus elevation-dependent weighting was used in this study.

**Table 1.** Summary of the GNSS data processing strategy for PPP.

| Item | Description |
| --- | --- |
| Observations | Ionosphere-free pseudorange and carrier phase measurements: GPS L1/L2, GLONASS L1/L2, Galileo E1/E5a and BeiDou B1I/B3I |
| Elevation mask | $7°$ |
| Weighting | Elevation-dependent: $4\sin(e)^2$ for elevations lower than $30°$; otherwise unit |
| Station position | Static |
| Receiver clock | Individual clock parameters for each GNSS constellation |
| Troposphere | Saastamoinen and GPT/GMF as a priori model (Saastamoinen, 1972; Böhm et al., 2006, 2007); remaining zenith wet delays are estimated as random walk parameters with a process noise of $10^{-9}\,m^2/s$ |
| Phase ambiguities | Float |
| Products | Precise satellite orbit and clock products from Center for Orbit Determination in Europe (Schaer et al., 2021) |
| Antenna model | IGS atx files (Rothacher and Schmid, 2010) for satellites; no correction applied for smartphones |
| Phase wind-up | Corrected (Wu et al., 1992) |
| Tides | Solid Earth tides, ocean tidal loading and pole tide (Petit et al., 2010) |

## 3 Data

In this study, two data sets were utilized to investigate PPP-based ZTD estimation using smartphones: (i) GNSS observations collected with a Google Pixel 4XL smartphone during a dedicated experiment located on the ETH campus rooftop and (ii) high-quality smartphone data crowdsourced from Germany. The subsequent subsections are dedicated to the characteristics of the acquired data sets and the description of the external data used for validation of the acquired smartphone-based ZTD estimates.

### 3.1 ETH rooftop experiment

Crowdsourced data are often not collected in favorable observation environments, and the availability of nearby geodetic-grade stations for ZTD evaluation cannot always be guaranteed. To explore the potential of smartphone GNSS data collected in open-sky environments for tropospheric monitoring, a Google Pixel 4XL smartphone was employed to collect 24 h of data on May 18$^{th}$ 2023. The smartphone was placed on the rooftop of the HPV building at the ETH Hönggerberg campus (Figure 2). It was capable of tracking GPS, Galileo, GLONASS and Beidou signals (Stauffer et al., 2023). The CAMALIOT app (See et al., 2023) was used to record raw GNSS data outputted by the embedded GNSS chip and antenna. The data were then converted to



RINEX files within the app so that they could be used for further analysis. It is worth mentioning that the CAMALIOT app was optimized for RINEX conversion, with a focus on ensuring receiver clock consistency between the pseudorange and carrier
phase measurements (Wang et al., 2021; Zangenehnejad et al., 2023). In addition, a patch antenna (ANN-MB-00), connected to a u-blox ZED-F9P receiver, was placed approximately 2 m away from the smartphone. This antenna-receiver combination represents a typical low-cost GNSS device (Hohensinn et al., 2022) and is suitable for performance comparison. The u-blox ZED-F9P receiver can track dual-frequency data from the GPS, Galileo and GLONASS constellations. For ZTD determination, the geodetic-grade station ETH2, located on the same rooftop (as shown in Figure 2), served as the benchmark.

Although the Pixel 4XL can record L5/E5a measurements for GPS and Galileo, respectively, the quality of these measurements is lower compared to those on the L1/E1 frequency (Stauffer et al., 2023). To enable high-precision PPP processing for the Pixel 4XL, the original L5/E5a measurements were not used. Instead, seven stations from the Automated GNSS Network for Switzerland (AGNES) network, situated at a distance of around 50 km from the smartphone, were employed to interpolate ionospheric delays and recover the measurements on the second frequency (Deng et al., 2009).

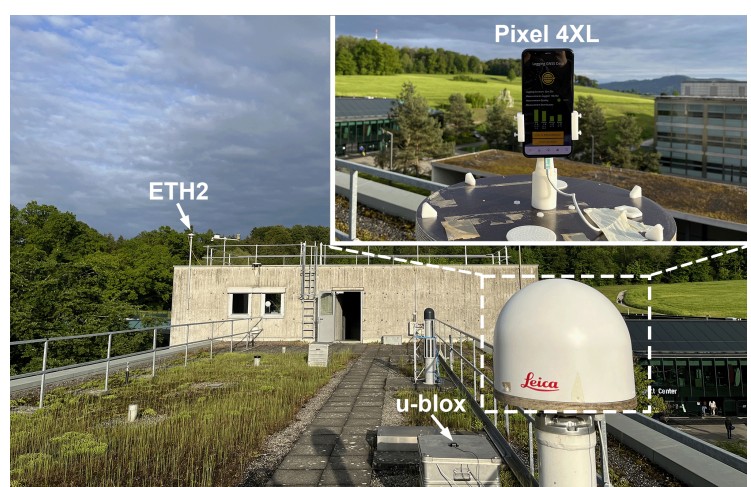

**Figure 2.** Configuration of the GNSS data collection experiment carried out on the rooftop of ETH campus. The Pixel 4XL smartphone was shielded by a weather-resistant radome. A u-blox device was employed for performance comparison, while the geodetic-grade station ETH2 served as the benchmark for ZTD estimation. The insert in the upper-right corner displays the Pixel 4XL under the radome.

## 3.2   Crowdsourced smartphone GNSS data

The smartphone GNSS data from Germany were used in this study to demonstrate the feasibility of ZTD determination based on the crowdsourced observations. From March 2022 to mid-May 2022, more than twenty thousand RINEX files were collected from Germany, comprising 21.0% of the total observation files collected worldwide. Figure 4 provides an overview of the data quality, focusing on three important quality indicators: $C/N_0$, observation duration and the presence of dual-frequency data.
Generally, higher $C/N_0$ values correspond to better data quality, and longer observation duration favors the ZTD estimation. The mean $C/N_0$ value is 27.1 dB-Hz, indicating that a significant fraction of the data were collected indoors. On average, the





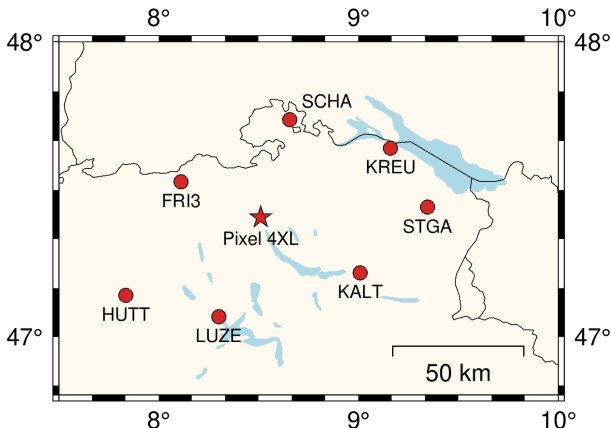

**Figure 3.** Distribution of AGNES stations utilized for the computation of ionospheric delays. The seven AGNES stations are indicated by red circles, while the red star denotes the location of the Pixel 4XL smartphone.

observation duration amounts to 3.5 h. Interestingly, $C/N_0$ tends to be lower for longer observation sessions, suggesting that longer data collection sessions are more likely to be conducted indoors. If we establish a criteria where $C/N_0$ is greater than 35 dB-Hz and the observation duration exceeds 0.5 hours, only 2.8% of the data meet the requirements and hold the potential

for ZTD estimation. Note that this criteria is empirically determined to show the data quality distribution and was not applied for practical data selection. Although more than 94.6% of the data included multi-GNSS observations, only 12.7% of the data contained dual-frequency measurements. Moreover, there were typically only 2-5 measurements available per epoch on the L5/E5a frequency, which is insufficient for PPP processing.

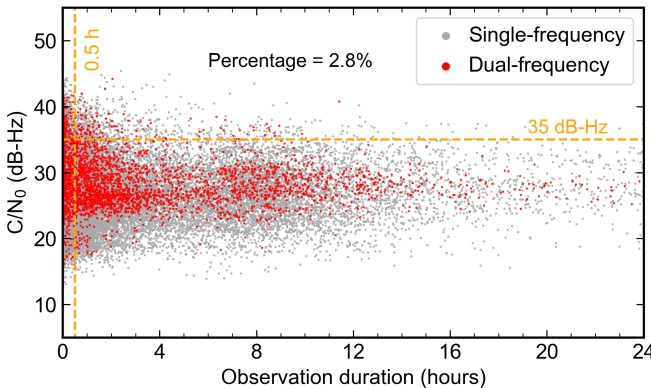

**Figure 4.** The distribution of $C/N_0$ and observation duration of the crowdsourced GNSS data in Germany. Each dot represents a RINEX file containing GNSS observations from a single session. Black dots indicate files with single-frequency observations only, while red dots represent files with dual-frequency observations. The horizontal dashed line denotes the $C/N_0$ threshold of 35 dB-Hz, while the vertical line denotes the observation duration threshold of 0.5 h.



As previously mentioned, to select high-quality data for tropospheric monitoring, the crowdsourced smartphone data initially
went through a classification process utilizing the established ML-based model (Figure 1). In this stage, the ML model identi-
fied 80 observation sessions from 20 users as 'Good'. Those 80 sessions were then processed with PPP for a positioning test,
with ionospheric delays being interpolated using the surrounding SAPOS (Satellite Positioning Service of the German State
Surveying) stations. Only data collected in static scenarios, exhibiting reasonable positioning precision, were further analyzed
for ZTD estimation. Finally, 20 sessions from 10 smartphone users showed high-precision positioning results and therefore
were employed for ZTD estimation. Figure 5 shows the distribution of these 10 smartphones in Germany and the correspond-
ing SAPOS stations utilized for ionospheric delay interpolation. To distinguish between these 20 observation sessions, each
smartphone was designated a unique character (A-J), and individual data collection sessions for the same smartphone were
identified by assigned numbers.

Given the arbitrary distribution of the crowdsourced data, nearby geodetic-grade stations that could serve as ZTD references
were not always available. To address this issue, we employed ERA5 products, publicly available from the European Centre
for Medium-Range Weather Forecasts (ECMWF), to compute ZTDs at the smartphone locations. We acquired ERA5 hourly
data on 37 pressure levels for Germany, with a spatial resolution of 0.25° by 0.25°. In addition, we leveraged an ML-based
tropospheric delay product, provided by the Chair of Space Geodesy at ETH Zurich via its Geodetic Prediction Center (Soja
et al., 2022), as an additional reference. It is worth noting that this ML-based product has been reported to achieve a global
accuracy of around 8 mm when compared to the ZTDs derived from GNSS observations (Crocetti et al., 2023).

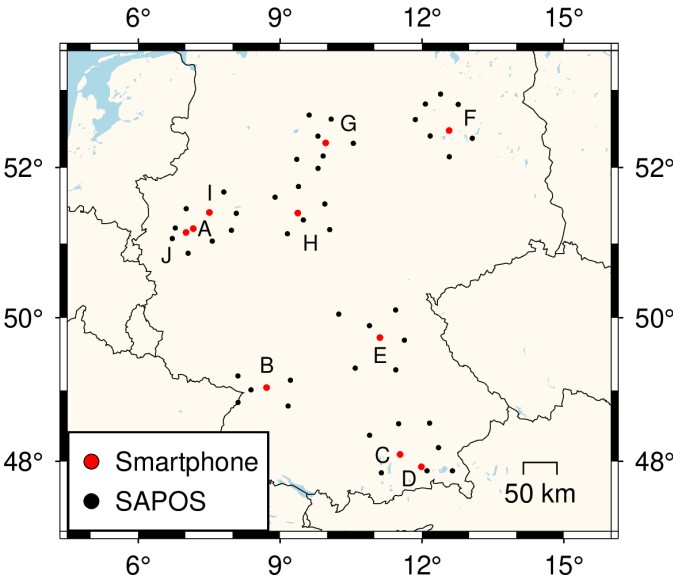

**Figure 5.** Distribution of selected high-quality crowdsourced smartphone data in Germany. The red dots represent smartphone locations,
while the black dots denote the employed SAPOS stations for the computation of ionospheric delays.





## 4    Results and discussion

The results of the ETH rooftop experiment are first presented in this section, focusing on data quality analysis and ZTD estimation. The ZTDs derived from the crowdsourced data in Germany are then introduced, complemented with their validation based on external reference ZTD products.

### 4.1    ETH rooftop experiment analysis

In this subsection, the GNSS data quality for the Pixel 4XL smartphone, u-blox and ETH2 are first presented. Then, the ZTD estimates derived from the Pixel 4XL and u-blox are evaluated, using ZTDs from ETH2 as references. Note that the Pixel 4XL and u-blox are abbreviated as PIXL and UBLX, respectively, in the following text.

#### 4.1.1    Data quality analysis

High-quality GNSS observations as well as the number of available measurements per epoch are essential factors for deriving accurate ZTD estimates. The observation environment on the rooftop of the ETH campus can be considered favorable, and therefore, one anticipates high-quality data from PIXL. As indicated in Table 2, the mean $C/N_0$ for PIXL is 41.9 dB-Hz, significantly exceeding the values characterizing most of the crowdsourced smartphone data. Nevertheless, it remains approximately 3 dB-Hz lower than the values recorded by ETH2 and UBLX, which can be explained by the polarization mismatch between the linearly-polarized smartphone GNSS antenna and the right-hand circular polarization of the GNSS signals (Zhang et al., 2013). Importantly, no significant differences are observed in satellite tracking performance among the three different devices, with roughly 9 GPS, 7 Galileo and 7 GLONASS satellites usable at each epoch. It is worth noting that the UBLX cannot track Beidou signals, and only three Beidou satellites were observed by PIXL. Consequently, observations from Beidou satellites were excluded from the ZTD estimation for this data set. With observations from GPS, Galileo and GLONASS, the mean PDOP values are 1.02, 1.11, and 1.07 for ETH2, UBLX and PIXL, respectively.

Observation noise serves as a straightforward metric of data quality. We calculated observation noises using time-differenced pseudorange and carrier phase measurements (Colosimo et al., 2011). ETH2 and UBLX exhibited similar pseudorange noises amounting to 0.17 m and 0.24 m, respectively. In contrast, PIXL exhibited a considerably higher pseudorange noise of 3.52 m. However, the noise level of carrier phase measurements was consistent among these three devices, with 0.003 m, 0.004 m and 0.004 m for ETH2, UBLX and PIXL, respectively. This observation aligns with findings from other studies, suggesting that smartphones can provide precise carrier phase measurements but tend to deliver less accurate pseudorange measurements (Zhang et al., 2018; Li and Geng, 2019). This is a promising feature for smartphone-based ZTD estimation, as carrier phase measurements are crucial for high-precision demanding applications.

#### 4.1.2    Evaluation of zenith total delays

We processed multi-GNSS data from PIXL, UBLX and ETH2 using PPP in static mode, as detailed in Table 1. Note that only GPS, Galileo and GLONASS observations were utilized, as UBLX lacked the capability to track Beidou signals. The ZTDs





**Table 2.** Data quality statistics for the GNSS observations collected by ETH2, u-blox and Pixel 4XL.

| Device | $C/N_0$ (dB-Hz) | PDOP | Pseudorange noise (m) | Carrier phase noise (m) |
|--------|-----------------|------|-----------------------|-------------------------|
| ETH2 | 44.2 | 1.02 | 0.17 | 0.003 |
| UBLX | 44.6 | 1.11 | 0.24 | 0.004 |
| PIXL | 41.9 | 1.07 | 3.52 | 0.004 |

derived from ETH2 were used as benchmark. Typically, ZTDs derived from geodetic-grade receivers can achieve a high level of accuracy, often within a few millimeters (Li et al., 2015; Wilgan et al., 2022). As shown in Figure 6, the ZTDs derived from UBLX exhibit a good agreement with those from ETH2, with an RMS value of 1.9 mm. This is noteworthy, given that UBLX is a low-cost device. Although antenna PCV errors were not corrected for UBLX, the ZTD bias is only -1.4 mm. In contrast, the ZTDs derived from PIXL show a much larger bias, approximately 6.0 mm. This bias is most likely attributed to the uncorrected PCV errors, considering the similar elevation-dependent patterns of PCV and ZTD. Currently, there is no available antenna PCV information for smartphones. However, there is a potential for smartphone manufacturers to publish such information for each smartphone model using the GnssAntennaInfo API provided by the Android 11 system (Google). Based on the ZTD estimates for the entire 24-h period, the resulting RMS value is 6.5 mm, which can be considered sufficiently accurate for tropospheric monitoring. The largest deviation from the reference time series (ETH2) is observed around 12 o'clock (GPST), which can be explained by the increased interpolation error of ionospheric delays near local noon.

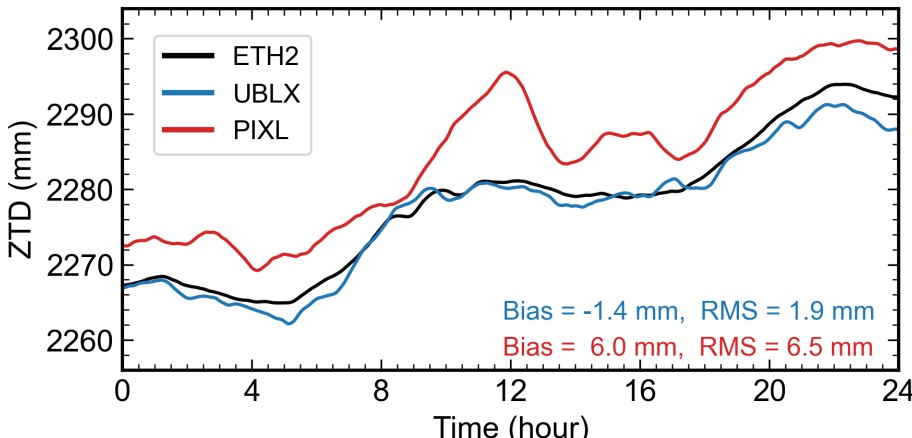

**Figure 6.** ZTD estimation using GNSS data collected on the rooftop of ETH campus. ZTDs derived from ETH2, UBLX and PIXL are represented by black, blue and red lines, respectively. The bias and RMS of the ZTD estimates with respect to ETH2 are provided in the bottom-right corner for UBLX and PIXL. Time is expressed as GPS Time (GPST).

In contrast to Continuously Operating Reference Stations (CORS), crowdsourced smartphone GNSS data are typically characterized by a short observation period, often spanning only a few hours or even minutes. A certain observing time span is




necessary to accurately separate ZTDs from other solve-for parameters, particularly the up component of station coordinates and receiver clock bias, during the parameter estimation process. To assess the impact of observation time span on ZTD estimation precision, we divided the 24-h data collected with PIXL into various time spans, including 5 min, 10 min, 15 min, 20 min, 30 min, 45 min, 1 hour, 1.5 hours, 2 hours and 3 hours. For each time span, we corrected the ZTD estimates for the 6.0 mm bias, and then computed the mean accuracy in comparison to the reference ZTD estimates acquired with ETH2.

The results are shown in Figure 7. It is evident that the mean ZTD estimation accuracy improves with longer observation duration. Interestingly, the benefits of extending the observation time span become less pronounced as the duration increases. This suggests that more GNSS observations have the greatest impact on ZTD estimation when the overall observation session is short. Notably, the mean accuracy is better than 10 mm when the time span is longer than 30 min. This is a promising finding, as collecting 30-min of GNSS data in an open-sky environment is very feasible within the common smartphone battery life.

Thus, a 30-min observation duration represents a reasonable trade-off between ZTD estimation accuracy and data collection efforts if the smartphone employed can track multi-GNSS signals.

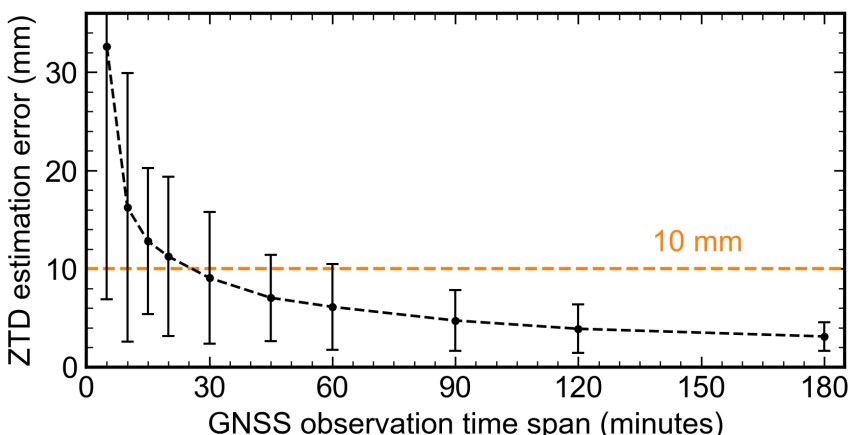

**Figure 7.** ZTD estimation accuracy using the PIXL data with varying observation time span. The black dots depict the mean error of the ZTD estimates with respect to ETH2. The one-sigma uncertainty is indicated by the error bar edges. Note that the ZTD estimates were based on multi-GNSS observations (GPS+Galileo+GLONASS) and the bias with respect to ETH2 was corrected.

To further investigate the influence of multi-GNSS observations on smartphone-based ZTD estimation with short observation time spans, we conducted an analysis in relation to three scenarios: GPS-only solutions, GPS+Galileo solutions and GPS+Galileo+GLONASS solutions. The results are summarized in Figure 8. Incorporating Galileo observations tends to im-

prove the accuracy of the ZTD estimates, especially for short observation sessions. For instance, when the observation session spans 30 min, the accuracy improves from 13.4 mm to 9.5 mm compared to the case where only GPS observations are utilized. On the other hand, the impact of GLONASS observations on ZTD estimation is less evident. Adding GLONASS does not always improve the ZTD estimation accuracy. This variability may be attributed to the inferior quality of the GLONASS observations collected by smartphones, as suggested in previous studies (Wang et al., 2023; Tao et al., 2023). Overall, the





results indicate the advantages of smartphones capable of tracking multi-GNSS data for ZTD determination, especially when observation sessions are short. Encouragingly, 94.6% of crowdsourced smartphone data contain multi-GNSS observations. It is also worth noting that, when using 24 h of the PIXL data, there is no obvious difference (approximately 0.1 mm) between the ZTDs derived from GPS-only and multi-GNSS data (cf. Wilgan et al., 2022).

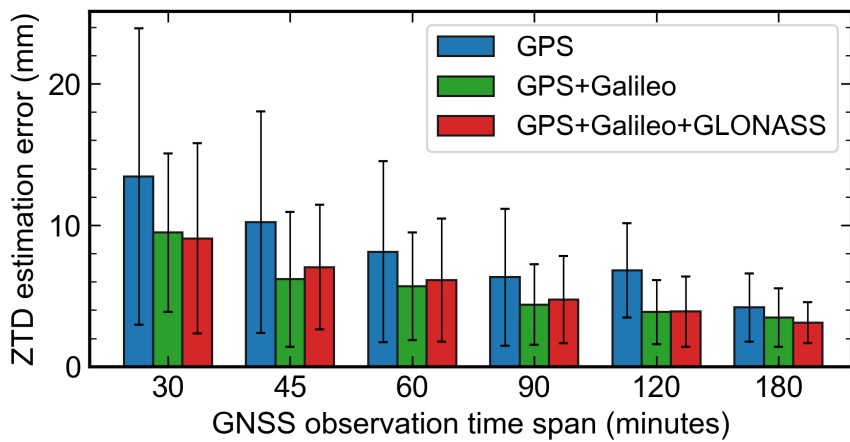

**Figure 8.** ZTD estimation error using the PIXL data with different GNSS constellation combinations and varying observation time spans. The bars depict the mean error of the ZTD estimates with respect to ETH2. The one-sigma uncertainty is represented by the error bar edges. Note that the bias with respect to ETH2 was corrected.

## 4.2 Crowdsourced data analysis

In this section, the data quality and the ZTD estimation performance of the crowdsourced smartphone data from Germany are presented.

### 4.2.1 Data quality analysis

Analyzing the quality of the crowdsourced smartphone data is beneficial for discovering the relationship between quality indicators and the accuracy of ZTD estimation. Figure 9 exhibits three important quality indicators, namely observation duration,
$C/N_0$ and PDOP, for each of the selected observation sessions (Figure 5). As discussed in Section 4.1.2, longer observation sessions are statistically linked to more accurate ZTD estimates. Notably, a 30-min duration represents a reasonable trade-off between the observation time span and the ZTD estimation accuracy. The average observation duration among the 20 selected sessions is 1.4 h, with 85% of them exceeding 30 min. It is intriguing to observe that the observation duration exhibits a user-dependent pattern, meaning that the same user tends to collect data of similar periods of time. For example, the three longest
observation sessions (H2, H3 and H4) were recorded by the same user and spanned between 4 and 5 h. In contrast, the shortest session, G1, spans a mere 16 min but records GNSS observations with a high mean $C/N_0$ value of 43.6 dB-Hz. The mean $C/N_0$ across all sessions is 40.0 dB-Hz. However, it is important to note that $C/N_0$ values are device-specific (Bilich et al., 2007),





making direct comparisons between different smartphones rather difficult. In general, higher $C/N_0$ values indicate favorable data collection conditions, and data with $C/N_0$ values exceeding 35 dB-Hz are likely to be collected in open-sky environments.

The PDOP value, on the other hand, is a direct metric concerning the quantity of available satellites and their relative locations in the local sky. A lower PDOP value is associated with more visible GNSS satellites and a better geometric distribution. In our case, the mean PDOP across all sessions is 1.3, indicating good observation conditions overall. There is no noticeable correlation among observation duration, PDOP and $C/N_0$, reflecting the high variability and complexity of crowdsourced smartphone data. It is worth mentioning that across the selected data sets, the noise level of carrier phase measurements is around 4 mm,

which is a promising factor for obtaining accurate ZTD estimates.

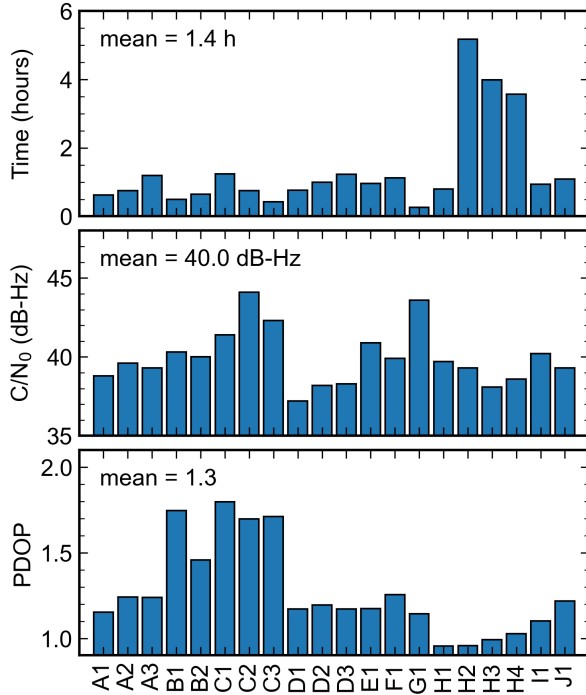

**Figure 9.** Data quality indicators for selected crowdsourced smartphone GNSS data from Germany. The characters are used to indicate different users and the numbers represent the different observation sessions from the same user.

### 4.2.2 Evaluation of zenith total delays

Figure 10 presents the ZTD estimation errors for the 20 selected high-quality data sets (Figure 5) in comparison with ERA5 and the aforementioned ML-based ZTD products. The mean RMS values, calculated based upon all considered sessions, are 26.9 mm and 26.4 mm when ERA5- and ML-based ZTDs are used as benchmarks, respectively. Notably, there is no

discernible distinction in the error patterns between the top and bottom panels of Figure 10, indicating a good agreement between the ERA5-based ZTDs and those derived from the ML-based products. Despite the data being crowdsourced, seven





data sets achieve an accuracy of better than 10 mm regarding ZTD estimation. These high-performing observation sessions are D1, D2, G1, H2, H3, I1 and J1, with mean RMS values of 6.3 mm and 3.9 mm when compared with ERA5 and ML-based products, respectively. It is also interesting to note that ZTD estimation accuracy tends to be user-dependent, with data from certain users consistently yielding more accurate results, as exemplified by user H. This could have been useful feedback for CAMALIOT app users, potentially motivating them to contribute more high-quality data during the crowdsourcing campaign. When analyzing the ZTD estimation error (Figure 10) and the data quality (Figure 9) together, data with lower PDOP values are more promising to yield more accurate ZTD estimates. This trend is especially noticeable in the data from users D-J, where the mean PDOP value for all the sessions that they uploaded is 1.11, and the mean RMS values for the ZTD estimates are 13.5 mm and 12.3 mm when compared to ERA5 and ML, respectively. Furthermore, longer observation duration and higher $C/N_0$ tend to contribute to improved accuracy of ZTD estimates. For example, observation sessions H2, H3, H4 span over 4 hours each, resulting in a mean RMS of around 6 mm. Another interesting example is session G1, which lasted only 16 minutes but had a mean $C/N_0$ value of 43.6 dB-Hz, leading to a ZTD estimation accuracy of 5.8 mm and 0.3 mm in comparison with the ERA5 and ML-based products, respectively. This highlights the potential of smartphone data collected in open-sky environments, even with a short observation duration, to make a contribution to tropospheric monitoring. It does not conflict with the finding summarized in Figure 7. Smartphone GNSS data characterized by short time span could still yield accurate ZTD estimates, albeit with a smaller probability.

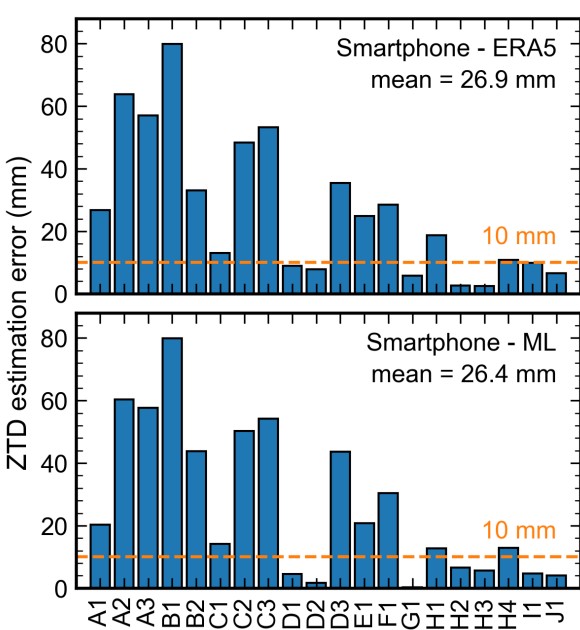

**Figure 10.** ZTD estimation errors using the crowdsourced smartphone data in Germany. The top panel shows the statistics where ERA5-based ZTDs are used as the benchmark, while the bottom panel uses ML-based ZTD time series as the reference. In both cases, the mean RMS error is denoted in the upper-right corner. The orange horizontal lines denote a threshold of 10 mm.




Figure 11 shows the ZTD estimates derived from session H2, alongside the ZTD time series derived from the ERA5 and ML-based products. This observation session is characterized with a duration of 5.2 h and observations to 27.5 GNSS satellites on average, including 8.9 GPS, 4.8 Galileo, 6.0 GLONASS and 7.8 Beidou satellites. The mean $C/N_0$ is 39.3 dB-Hz, indicating that these measurements were likely collected in open-sky conditions. It is worth noting that the original smartphone observations are limited to single-frequency. A noticeable ascending trend, approximately 10 mm over 5 h, can be observed in all three ZTD time series. The ZTDs derived from smartphone data exhibit a closer agreement with those from ERA5, with an RMS of 2.7 mm. A slightly larger bias is observed between the smartphone-based ZTDs and the ML-based ZTD product, with a bias of 6.4 mm. However, it cannot be concluded that the ML-based ZTDs are inferior to the other two. This is primarily because the bias between the smartphone-based ZTDs and the ERA5- or ML-based ZTDs could be offset or magnified by the bias resulting from uncorrected PCV errors for the smartphone during PPP processing. Nevertheless, it demonstrates that ZTD estimation with an accuracy better than 10 mm can be achieved with crowdsourced smartphone GNSS data, underlining their potential for accurate tropospheric monitoring.

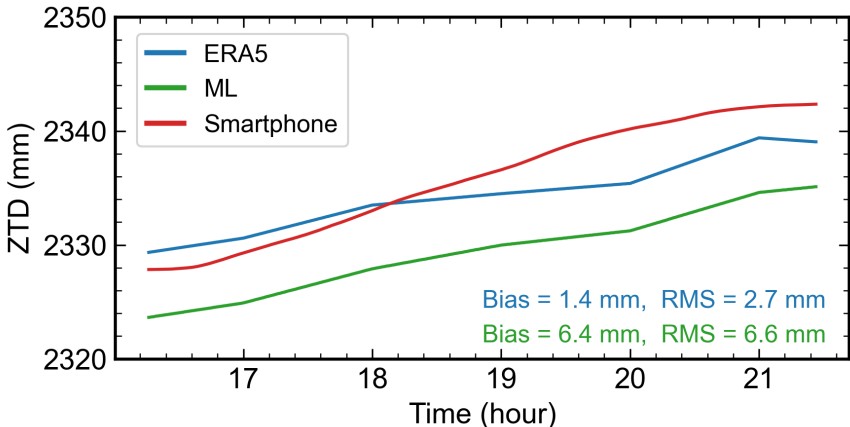

**Figure 11.** ZTD estimates from one crowdsourced smartphone GNSS data set (H2) in Germany. The ZTDs derived from ERA5, an ML-based ZTD product and smartphone GNSS data are represented by blue, green and red lines, respectively. The bias and RMS of the smartphone-based ZTDs with respect to ERA5- and the ML-based ZTDs are listed in the bottom-right corner.

## 5 Conclusions

Crowdsourced smartphone GNSS data have the potential to densify existing geodetic-grade GNSS networks, providing valuable observation sources for GNSS meteorology. This study represents the first successful demonstration of achieving high-precision ZTD estimation using crowdsourced smartphone GNSS data. We harnessed GNSS data collected in Germany during the CAMALIOT crowdsourcing campaign to demonstrate the feasibility of smartphone-based ZTD estimation and investigate the quality of the ZTD estimates that such type of GNSS data can provide. The dedicated data processing pipeline was introduced, including data selection and ZTD estimation using PPP. Ionospheric delays at smartphone locations were interpolated



using data from surrounding geodetic-grade GNSS stations and then the second-frequency observations were recovered using the SEID method for ionosphere-free PPP. This approach overcomes the issue concerning the limited availability and poor quality of usable dual-frequency data from most smartphones.

To validate our data processing method and to gain insights into smartphone-based ZTD estimation, a dedicated 24-h data collection experiment was conducted on the rooftop of the ETH campus. The performance of a Google Pixel 4XL smartphone was evaluated alongside a u-blox GNSS receiver in terms of ZTD estimation. Compared to the ZTDs derived from a geodetic-grade receiver located at the same rooftop, a ZTD estimation accuracy of 6.5 mm was achieved for the smartphone data when incorporating external ionospheric information. The u-blox receiver, with its own dual-frequency observations, reached an

accuracy of 1.9 mm. Based on the performed analysis, it was observed that multi-GNSS observations, especially from Galileo, improved the ZTD estimation accuracy for data sets characterized by short time spans. Our investigation also demonstrated that a 30-min observation duration served as a reasonable trade-off between the data collection effort and the ZTD estimation accuracy. With 30 min of multi-GNSS observations from the Pixel 4XL smartphone, we can achieve a mean ZTD estimation error of less than 10 mm. This finding can serve as a valuable insight concerning future smartphone GNSS data crowdsourcing

campaigns.

Twenty high-quality data sets crowdsourced from volunteers in Germany were processed with the same method to obtain ZTD estimates. Compared to the ZTD benchmarks derived from ERA5 data and an ML-based ZTD product, it was shown that a mean accuracy of about 26 mm could be achieved. A comprehensive analysis of data quality indicators and ZTD estimation accuracy indicates that data sets with smaller PDOP values tend to yield more accurate ZTD estimates. Moreover, longer

observation durations and higher $C/N_0$ values can be also helpful to identify data sets that can result in accurate ZTD estimates.

While we have demonstrated that high-precision ZTD determination can be achieved with crowdsourced smartphone GNSS data, certain limitations remain. Our current method relies on the interpolation of ionospheric delays from surrounding geodetic-grade stations, and we have primarily explored smartphone data collected in static scenarios. Future research could explore the use of original dual-frequency observations from capable smartphones and data collected on kinematic platforms, such as vehi-

cles. Additionally, antenna PCV corrections were not applied for smartphone GNSS data processing in this study. The potential benefit of antenna calibration for ZTD estimation could also be investigated in future studies. In conclusion, this study demonstrates that, with careful selection and processing, crowdsourced smartphone GNSS data can produce high-precision ZTD estimates and potentially benefit tropospheric monitoring and weather forecasting, especially as embedded GNSS antennas and chips continue to improve in the future.

*Data availability.* The GNSS data collected on the rooftop of ETH campus is available from the corresponding author upon reasonable request. The crowdsourced smartphone data currently cannot be shared due to privacy concerns. The ERA5 grids and the ML-based tropospheric delay products can be accessed from https://cds.climate.copernicus.eu and https://gpc.ethz.ch/Troposphere/, respectively.



## Appendix A: ML-based smartphone data selection

We employed the Random Forest model to develop a classifier for smartphone data from the CAMALIOT crowdsourcing
campaign. A subset of the crowdsourced data served for training and testing the model. Given the potential fluctuation in data
quality during an observation session, we segmented the original GNSS measurements into hourly RINEX files. Only segments
labeled as 'Good' were spliced to produce a consolidated RINEX file. If any hourly segments were labeled 'Good', the entire
original file was regarded as 'Good' and the consolidated file would be used for further processing. We then extracted a set
of epoch-wise quality indicators from each hourly segment and their time series were visualized for manual labeling (e.g.,
Figure A1). The corresponding mean and standard deviation (STD) values were calculated and utilized as input features
for the ML-based classifier (Table A1). Note that a threshold-based classifier would be overly intricate considering the high
complexity among different quality indicators. Nonetheless, we initiated the labeling process by applying threshold criteria to
flag blatant low-quality segments. Specifically, segments with a mean $C/N_0$ below 30 dB-Hz or carrier phase noise exceeding
0.1 m were labeled 'Bad'. The remaining data underwent manual labeling based on our expertise with high-quality smartphone
data collected in open-sky environments. Finally, the training data set comprised 1,700 'Good' segments and 3,400 randomly
sampled 'Bad' segments. We evaluated the developed ML-based classifier on a test data set, consisting of 425 'Good' segments
and 1,792 'Bad' segments. The classifier exhibited precision and recall scores of 0.96 and 0.97, respectively (Table A2),
meeting the requirements for the CAMALIOT campaign's operational data classification task.

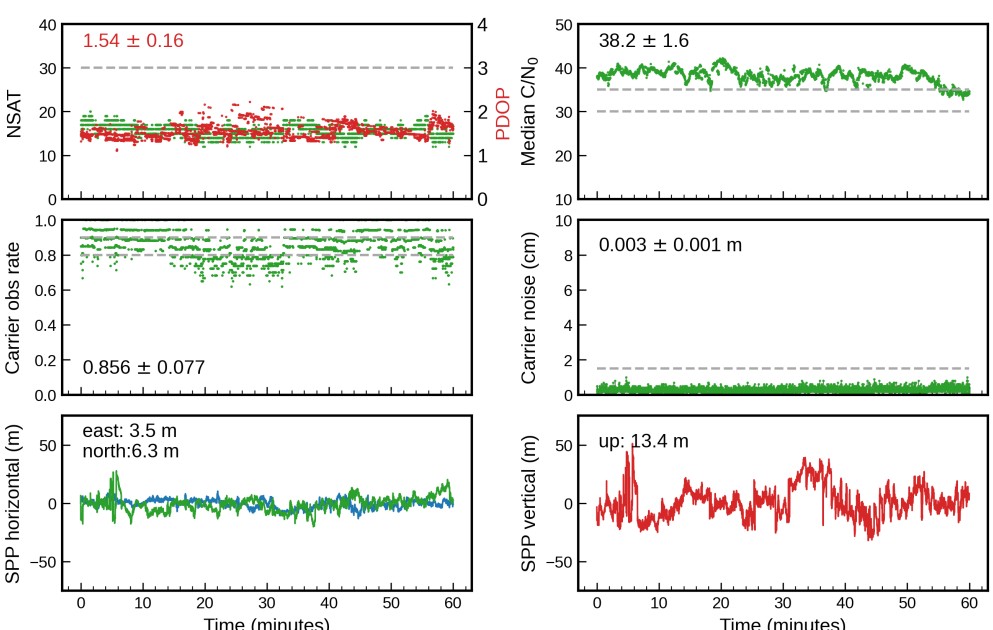

**Figure A1.** Visualization of several epoch-wise quality indicators for manual labeling of crowdsourced smartphone data.



**Table A1.** Input features for training and testing the ML-based smartphone data classifier.

| Features |
| --- |
| Number of epochs |
| Mean number of visible satellites |
| Mean PDOP and its STD |
| Mean C/N$_0$ and its STD |
| Percentage of available carrier phase observations |
| Mean carrier phase noise and its STD |
| SPP precisions in the east, north and up components |

**Table A2.** Evaluation of the developed ML model for crowdsourced smartphone data classification on a test data set. (TP: true positive, TN: true negnative, FP: false positive, FN: false negative, $precision = \frac{TP}{TP+FP}$, $recall = \frac{TP}{TP+FN}$, $F1 = 2 \cdot \frac{precision \cdot recall}{precision+recall}$)

| TP | TN | FP | FN | Precision | Recall | F1 |
| --- | --- | --- | --- | --- | --- | --- |
| 410 | 1775 | 17 | 15 | 0.96 | 0.97 | 0.96 |

*Author contributions.* YP analyzed the data, visualized the results and wrote the manuscript. GK, RW, TS, LS, IM, VN and BS contributed to
the implementation and organization of the CAMALIOT crowdsourcing campaign. LC provided the ML-based tropospheric delay products.
GD provided the GNSS data from the SAPOS network. GM helped to compute zenith total delays from the ERA5 grids. BS, LS, GK, MR,
IM acquired the funding. BS supervised the study. All authors reviewed the manuscript and approved it for publication.

*Competing interests.* The authors declare no competing interests.

*Acknowledgements.* The authors would like to acknowledge Swisstopo for the access to the AGNES GNSS data, the IGS analysis centers
for providing the high-precision GNSS satellites products, and all the CAMALIOT app users for their participation in the crowdsourcing
campaign. This work has been funded by the ESA NAVISP Element 1 Program (NAVISP-EL1-038.2).



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
