# Peer review of "Determination of High-Precision Tropospheric Delays Using Crowdsourced Smartphone GNSS Data"

_EGUsphere, 2024_

## Author Comment (AC1)

Reviewer #1

GENERAL COMMENTS

This is a very interesting manuscript that deserves publication on the potential of using ZTDs derived from raw GNSS data obtained from private citizens smartphones. Supplemented with an example of ZTD data from a smartphone providing GNSS data in an idealized setting and compared to ZTD derived from a professional grade GNSS receiver.
I congratulate the authors of having written a nice, easy to read manuscript almost ready to publish.

Thank you for recognizing the value of our manuscript!

Compared to previous tests of usage of other crowsourced smartphone data, such as for example pressure (see e.g. Hintz et al, https://doi.org/10.1002/met.1805) it is clear that for derivation of ZTD from raw GNSS smartphone data much longer un-interrupted time-series are required for the mobile phone data to be useful. On top there is a benefit from obtaining the GNSS data, when the phone in an open environment.
It would be interesting to include in the article information about:
1. The local time of day distribution of the full data volume versus the volume kept for analysis after filtering.

Thank you for the comment! The figure below shows the distribution of the local time when users started collecting GNSS data in Germany. It can be concluded that most data were collected during the daytime or in the late evening, especially for the selected data. This figure is not included in the revised manuscript since it is not closely related to the main topic. However, a sentence has been added to section "3.2 Crowdsourced smartphone GNSS data" to mention this finding:
"*Most of the data were collected during the daytime or in the late evening.*"

[Figure]

Figure 1 Distribution of local time when users started collecting data in Germany. The upper panel shows the distribution of all the crowdsourced data and the lower panel shows the results of the 20 selected data sets.

2. Were the volunteers providing the data given any information about how to use (e.g., how long in the proper mode) or place (indoor/outdoor/sky view) or not move their devices prior to the experiment?

Yes, some basic instructions on how to collect GNSS data were given when users opened the CAMALIOT app for the first time. For example, users were told to leave their smartphones stationary in a place with an open-sky view. However, these requirements could not be enforced.

We added one sentence in section "2.1 Data selection" to include this information:
"*For example, users were advised to place their smartphones in a stationary position with an open-sky view.*"

3. Many Android phones contain also a pressure sensor, data from that could be collected simultanously, the two types of data potentially improving usage when used together (just your thoughts on this).

Thanks for the comment! Yes, it could be beneficial to collect air pressure measurements along with raw GNSS data. For example, (1) precise locations could be derived from GNSS for barometric measurements and further correct barometric measurements with altitude information, (2) the zenith wet delays can be precisely separated from GNSS-derived zenith total delays if air pressure and temperature are available.

Actually, this functionality has been implemented in the latest version of the CAMALIOT app. Users can now choose to collect and upload measurements from environmental sensors, such as air pressure and illumination, along with GNSS observations. Hence, we have added a sentence in the outlook part of section "5 Conclusions":
"*The latest version of the CAMALIOT app allows users to record environmental sensor measurements, such as air pressure, which could further contribute to meteorological applications.*"

SPECIFIC COMMENTS

The smartphone based curves in figure 11 appear to me surprisingly smooth. Is that due to constraints in the data processing of the raw GNSS data or subsequent smoothing of the ZTDs?

Yes, a between-epoch constraint (i.e., process noise) was applied to the ZTD estimation, which is specified in Table 1 in the manuscript. No subsequent smoothing was applied.

---

## Author Comment (AC2)

Reviewer #2

Tropospheric delay information is valuable to both spatial geodetic measuring and weather forecasting. Ground-based GNSS has advantages of high-accuracy, high temporal resolution and can be operated under all weather conditions. However, the number of traditional geodetic GNSS receivers is limited due to the cost. The use of crowdsourced smartphone GNSS data is promising to provide much denser tropospheric delay information due to the massive smartphone users. This paper carried out novel investigations on the ZTD estimation based on crowdsourced data collected from the CAMALIOT campaign. The paper is generally well organized and the experiments were well designed. But, in my opinion, there are still many work to do before you can say 'this is the first successful demonstration of high-precision ZTD determination with crowdsourced smartphone GNSS data'. The main limitation of this study is that the selected data for the ZTD estimation are all from static users in an open-sky environment. Can these data be called real crowdsourced data? As we know, most smartphone users are indoor or kinematic. So the results from open-sky static users are just too ideal. The data from the 10 crowdsourced users in this study are similar to the data collected by repeating 10 times of experiments at ETH in sect 3.1.

Thanks for the comments! Yes, we admit that there are still some limitations in this study, which were also mentioned in the last paragraph of section "5 Conclusions". Hence, we changed the sentence "This study marks the first successful demonstration of high-precision ZTD determination with crowdsourced smartphone GNSS data" to "*This study demonstrates high-precision ZTD determination with crowdsourced smartphone GNSS*" in the abstract.

Currently, we focus on static data for the following three reasons: (1) Most crowdsourced data from the CAMALIOT project were static, since the app users were advised to place their smartphones in a stationary position. (2) There might be some kinematic data collected by the users. However, the quality of kinematic data is usually lower, which makes them more difficult to process. (3) It is more challenging to evaluate ZTD estimates from kinematic data. Thus, static data were the focus of this manuscript and a future study would be needed to investigate kinematic data.

We do not agree that "Can these data be called real crowdsourced data?" and "The data from the 10 crowdsourced users in this study are similar to the data collected by repeating 10 times of experiments at ETH in sect 3.1". As you also mentioned, most crowdsourced data were collected indoors and of low quality. It is basically not feasible to derive reliable ZTD estimates from indoor GNSS data. However, this study demonstrates that data collected outdoors (not necessarily open-sky environments) from a crowdsourcing campaign could still exist and high-precision ZTD could be derived from them. The 20 high-quality data sets from the 10 users were selected with our dedicated data selection method. They were real

crowdsourced data and used for the demonstration of ZTD estimation. However, the data quality of these data sets was not necessarily the same as the one we collected at ETH, which can be observed in Figure 9 in the manuscript. Basically, two major differences can be found: (1) The time span of the crowdsourced data was much shorter (from 30 minutes to 5 hours), which makes it more challenging to derive precise ZTD estimates. (2) The observation environments of the crowdsourced data could be worse than the experiment conducted at ETH. For example, the C/N0 and PDOP values of some data sets were not so good, indicating they might be collected in less favorable environments.

Other minor comments include,

L56: why the accuracy of Benvenuto et al. (2021) is so bad with errors of several cm compared to several mm in other studies? Please make necessary comments here.

We think the main reasons are:
(1) The original dual-frequency GNSS data (L1+L5) from the smartphone were used by Benvenuto et al. used. Usually, the data quality on L5 is worse than L1 (Stauffer et al., 2023). In addition, a smaller number of GPS satellites are capable of transmitting L5 signals (about half of the GPS constellation), which leads to a worse observation geometry.
(2) Benvenuto et al. (2021) used publicly available software to process smartphone data. The software might not be refined for smartphone data processing. They also mentioned in their manuscript that software played a crucial role for precise ZTD estimation.

We modified the corresponding sentence in the revised manuscript:
"*Benvenuto et al. (2021) also explored ZTD estimation with original dual-frequency GNSS data of a Xiaomi 8 smartphone and publicly available software. They reported an accuracy of several centimeters*"

L132: the PCO corrections were not applied either, right?

That is right. PCO corrections were not applied.

L138: why the C/N0-dependent weighting tends to introduce artifacts in ZTD estimates in your results? Several studies have showed that the traditional elevation-dependent weighting strategy may not applicable for smartphone data. Why you still use this strategy?

Thanks for the comment! Actually, we had tested both weighting methods for smartphone GNSS data processing. The ZTD estimation results using different weighting methods are presented in the figure below. It is found that the fluctuation is larger if the C/N0-dependent weighting method is used, especially during the period 0-4 h and 20-24 h. That might be

attributed to the irregular gain pattern of smartphone GNSS antennas, which causes the artifact signals in ZTD estimates. Although the bias is a bit smaller when the C/N0-dependent weighting method is used, we think the overall signal is more important, since the bias is harder to interpret.

It is true that several studies adopt C/N0-dependent weighting method to achieve better positioning accuracy. However, the parameter of interest in this study is ZTD (instead of coordinates), whose estimation can be influenced by the choice of weighting methods. This has not been thoroughly investigated by the GNSS community. Thus, we still prefer the elevation-dependent weighting method for ZTD estimation based on our preliminary results. Drawing a more definitive conclusion would require another dedicated study in the future.

[Figure]

Figure 2 ZTD estimation using different weighting methods for PIXL data. The bias and STD of ZTD estimates with respect to ETH2 are provided in the bottom-right corner.

L231: why did you attribute the 6 mm bias to the uncorrected PCV errors? Can it also be possible due to the worse data quality in smartphone observations? You didn't apply PCV corrections for UBLX either, but the bias was much smaller.

Thanks for the comment! Yes, it could also be attributed to the worse quality of smartphone GNSS data. We modified the corresponding sentence in the revised manuscript:
"*This bias is* most *likely attributed to the* inferior *data quality from the smartphones and the uncorrected PCV errors, considering the similar elevation-dependent patterns of PCV and ZTD.*"

Figure 10: you need to give the number of ZTD samples in each case.

The figure was updated as suggested:

[Figure]

Figure 10 ZTD estimation errors using the crowdsourced smartphone data in Germany. The top panel shows the statistics where ERA5-based ZTDs are used as the benchmark, while the bottom panel uses ML-based ZTD time series as the reference. In both cases, the mean RMS error is denoted in the upper-right corner. The orange horizontal lines denote a threshold of 10 mm. The cyan line represents the number of ZTD samples used for statistics.